# Self-Assessment of Preparedness among Critical Care Trainees Transitioning from Fellowship to Practice

**DOI:** 10.3390/healthcare7020074

**Published:** 2019-05-30

**Authors:** Laura Watkins, Matthew DiMeglio, Krzysztof Laudanski

**Affiliations:** 1Department of Pediatrics/Critical Care, School of Medicine and Dentistry, University of Rochester Medical Center, 601 Elmwood Ave, Box 667, Rochester, NY 14642, USA; Laura_Watkins@urmc.rochester.edu; 2Philadelphia College of Osteopathic Medicine, Philadelphia, PA 19131, USA; matthewdi@pcom.edu; 3Department of Anesthesiology and Critical Care, Hospital of the University of Pennsylvania, Philadelphia, PA 19104, USA; 4Leonard Davis Institute of Health Economics, Philadelphia, PA 19104, USA

**Keywords:** graduate medical education, critical care, fellowship, employment

## Abstract

This study evaluated the subjective assessment of preparedness needs of critical care trainees and recent graduates between 2013 and 2014. A questionnaire was developed and validated by the subcommittee of the In-Training Section of Society of Critical Care Medicine (SCCM). The survey was deployed twice between December 2013 and January 2014 via email to any trainee or individual graduated from a critical care fellowship within the previous three years. Six percent (180) of all individuals completed the survey, and 67% of respondents had recently interviewed for a job. Northeast was the preferred location for a job (47%), and academia was favored over private practice (80% vs. 15%). Of the respondents that secured an interview, 55% felt prepared for the interview, 67% felt prepared to build an adequate job portfolio, 33% received formal guidance from their mentor/training program. 89% of total respondents agreed it is important to participate in a formal training course in job search, portfolio development, and interviewing process. The preferred sources of training were equally distributed between their home institution, webinars, and SCCM. There is an ongoing need in education regarding the transition period from fellowship to practice.

## 1. Introduction

The transition from being a fellow to an attending is a complicated process that presents unique challenges. The fellowship is the time to master the pathophysiology behind critical care delivery, to become competent in procedures, to develop research, evidence-based medicine, and presentations skills, and the time to start learning leadership, professionalism, and communication skills [1,2,3]. Often the proportion to which these competencies are developed depends on personal characteristics related to emotional intelligence, social communication, and interpersonal skills. In the last decade, training programs have begun to place more emphasis on teaching these skills using a variety of techniques and approaches [4,5,6,7,8,9]. However, little has been done to evaluate the effectiveness of these programs, which hope to address the issues, but may not succeed [10,11]. Furthermore, the changing market demands add stress, stretching the non-medical skills of fresh graduates in both academic and non-academic settings [12,13]. Dealing with inequalities in the process of job hiring remains problematic [14,15]. 

The transition period is the most sensitive in unveiling difficulties which derive from lack of appropriate preparation and acquired skills [16,17]. Conflict often originates from unmatched expectations [17]. Retention in the first job and a start to a successful career often depends on elements developed during this early period: Negotiation of an adequate contract, creating a good combination between the potential of the candidate and the needs of the institution, as well as securing early mentorship [2,5,12,13,18]. One step in understanding how the program continues to address these problems is to evaluate the perception of the critical care fellowship graduates on their preparedness for professional life.

Here, we investigated critical care fellows and junior critical care practitioner’s perception regarding their preparedness to transition to the first job demands and requirements. We specifically addressed the perception of support they have received from their training program as it relates to elements of resume building, portfolio creation, job search, interviewing process, and job decision. In contrast to other studies, we focus on junior critical care professionals (young attending, fellows, nurses, respiratory therapists, and pharmacists) [16]. Considering that several societies provide targeted courses and the Accreditation Council for Graduate Medical Education (ACGME) stresses the preparedness for transition to the first job, our study sought to evaluate the perceived preparedness of trainees and recent graduates for their first job transition.

## 2. Materials and Methods 

This study was designed to examine a cross-sectional cohort of critical care fellows and recent graduates. The Society of Critical Care Medicine (SCCM) was engaged to facilitate the dissemination of the survey. Before dissemination, the SCCM Research Committee and the Executive Committee approved the survey. Based on both institutional and SCCM guidelines, this study was deemed Institutional Review Board (IRB) exempt. 

The targeted population included: medical doctors (MD), doctors of osteopathic medicine, registered nurses, physician assistants, respiratory therapists, pharmacists, and residents currently in training or within 3 years from graduation. The database of SCCM membership was searched for subjects that met the study selection criteria. The SCCM database contains self-reported biographical information that users provide during the SCCM membership process. The survey was deployed twice via email containing links to the survey site during the period of December 2013 to January 2014. 

The survey was developed by members of the In-Training Section of SCCM after non-structured interviews with several fellows of critical care programs. A prior survey was used as a starting point to generate the questions [16]. The first version of the survey was adopted and circulated among members of the In-Training section. After modification of the survey, it was deployed to fellows participating in the program at the University of Pennsylvania. Two questions were modified as they were judged ambiguous by participants. 

The survey included demographics, desired job description, subjective sense of preparedness for job search, and interviewing readiness. While most of the specialties possessed a singular pathway to a critical care fellowship, multiple pathways currently exist for internal medicine. This includes the most common pathway: a three-year pulmonary/critical care fellowship, as well as a two-year critical care fellowship [19]. All the fellowship pathways within internal medicine were combined during analysis. The survey also inquired about interest in formal courses, preferred media of education, and possible topics of improvement. Most of the questions were in a multiple-choice format allowing more than one answer, except when specified. The questions regarding the subjective sense of preparedness for job search and interview followed a 5-point Likert scale. The participants were given an opportunity to express their opinion freely and offer suggestions at the end of the survey. A copy of the survey is attached as a Appendix A. Google Forms was used to collect the data.

The data were analyzed using SPSS (version 21, IBM) and is presented as percentages, or as median with the corresponding 25%–75% interquartile range (IQR) where appropriate. The Mann–Whitney U test was utilized to compare the distribution of intensive care specialties amongst medical doctors and other medical providers.

## 3. Results

Surveys were sent to 3000 individuals and 180 responses were obtained (response rate of 6%). A total of 160 responses were from physicians (89%) and 125 respondents (69%) were currently completing a fellowship. The distribution of specialties among the studied cohort is shown in Table 1. The majority of the respondents were physicians in training (Table 1). Of the respondents, 120 (66%) were interviewing currently or had interviewed within the past 4 years. Of these respondents, 68 (57%) of them were an MD or non-MD currently in training while the remaining were post training at the time of interview (Figure 1). The distribution of intensive care specialties was not statistically different amongst medical providers (MD and non-MD) in training.

The median number of locations interviewed was 3 (IQ_25%–75%_, 0–4), with a significant difference between people who were in training and those who had finished training (Figure 2). Respondents currently in training had a median of 2 (IQ_25%–75%_, 0–4), while those who had graduated had a median of 3 (IQ_25%–75%_, 2–5) (*p* = 0.036). The most common sources for job leads were: direct contact with the potential employer or recruitment agent (104, 87%), word of mouth (100, 83%), suggestion from mentor/fellowship director (87, 73%), and the SCCM site (58, 48%). Interestingly, journal advertising was a negligible source of job options. The desired job characteristics described by the entire cohort are as follows: 85 (47%) hoped to obtain a job location in Northeast US, with the rest of choices being equally distributed between Southeast, Northwest, Southwest, Northcentral (approximately 25% for each region) (Table 2). One hundred and forty-four (80%) respondents preferred an academic environment, and a similar percentage wanted a full-time job (Table 2). Interestingly, many of those who were in training responded that they were not currently interviewing (54, 43%).

The respondents who had interviewed in the past four years or were going to be interviewing (120, 66%), were further surveyed regarding the subjective sense of preparedness for job search and interview. The questions were posed in a 5-point Likert scale, however, for the ease of data description we aggregated the positive responses (defined as strongly agree and agree on the Likert scale), as well as the negative responses (defined as strongly disagree and disagree on the Likert scale). Of the respondents, 67% felt well prepared to create a portfolio for job interviews, however only 33% received formal guidance from the mentor/ training program in preparing their portfolio. Only 55% felt prepared for the interviewing process, with pediatric specialists feeling the least prepared and anesthesia specialists feeling the most prepared. The training program, including the mentor, facilitated resume building in only 42% of cases; with pediatrics, internal medicine, and anesthesia describing the most deficiencies. However, training programs and mentors guided or helped the trainee search for a position in 55% of cases (least in internal medicine, 38%), and helped in deciding which position to choose in 53% of cases (again, least in internal medicine, 29%).

A total of 160 (89%) respondents agreed that it is important to participate in a formal course on job search, portfolio development, and understanding the details of the interviewing process. Only 8% of respondents thought this course should be offered independently of training. This training was preferred at the beginning of the fellowship by 18%, toward the end by 31%, however, most respondents described a desire for this course in the middle or throughout training (40% and 43%, respectively). Courses offered by the home institution (73% of respondents) and webinars (71% of respondents) were the most commonly desired course format (Figure 3).

Regarding potential SCCM programs, the most desired topics consisted of the financial aspects of a first job (80.6%) followed by understanding career pathways and negotiating contracts (73.3% and 72.8%, respectively). Interestingly, maintaining work-life balance and mock job interviews were among the least desired topics (46.1% and 32.2%, respectively) (Figure 4A). Respondents felt that the most appropriate topics for SCCM programs were advancing within an academic setting and understanding different leadership styles (56.7% and 53.3%, respectively). The least appropriate topic was found to be dealing with a malpractice lawsuit (38.3%) (Figure 4B).

## 4. Discussion

Our study of graduating critical care fellows suggests that they feel unprepared for the interview process and that their programs have not supported them in building a competitive resume. Similar problems were reported in prior studies, but with the implementation of classes, courses, and support systems offered by residencies, fellowships, and professional societies, there was hope that pre-transition issues would be reduced [1,7,10,17,18]. The apparent lack of career development mentoring in our sample (37% of total respondents) confirms multiple previous studies [16,20,21,22,23]. A previous study among critical care physician-scientists found that fellows most commonly requested more information pertaining to hiring practices and job expectations [20]. A previous national survey of critical care practitioners in 2011 found that over half of respondents do not have a career development mentor in critical care [16]. 

Our findings also suggest that these graduates feel that their programs have supported them in searching and deciding on a position. However, this sentiment was not consistent throughout specialties, with internal medicine trainees feeling the least supported in developing their portfolio and searching for a position. This sentiment was affirmed by internal medicine trainees expressing the highest percentage of agreement in the utility of formal training in job searching and portfolio development.

Despite several measures to improve career development among critical care trainees, our study reveals that a gap persists between the needs of trainees and the programs delivered by training institutions. Our study also indicates that young attendings feel that their training programs should address non-clinical needs in a deeper way than currently delivered. This sentiment is echoed in the literature, which describes that several issues related to job satisfaction may stem from an unfit match between the practitioner’s expectations and the reality of their first job. It is suggested that this mismatch may stem from the lack of adequate preparation for job transition. This undue stress from lack of preparation may contribute to the burnout that is endemic among internal medicine and critical care professionals [24,25,26,27]. Difficulty in finding a mentor, or mentors failing to meet the needs of mentees, are among the causes of delayed entry of young investigators into independent National Institutes of Health (NIH) funding [28]. Frequent job switching may result from poor communication skills and mismatched expectations [29]. All these shortcomings lead to unfavorable clinical outcomes, and high attrition of the providers [21,30]. 

Our study has several limitations. The internal medicine pathways to critical care were under-represented in our study compared to national estimates of the intensivist workforce [31]. Although our response rate is similar to previous studies using the SCCM database, this suggests that our data may not be representative of the entire critical care workforce [16]. Moreover, responses could also be disproportionately from individuals that were struggling to find a job, indicating potential bias. Our survey also did not objectively address how fellow’s perception of preparedness in the transition to job correlates to success in that position. This offers several interesting avenues of future research and potential quality improvement projects for training programs to monitor the impact of educational efforts on career success among alumni.

A prior study addressed only trainees, but several common motifs are evident [16]: High desire to practice in the Northeastern United States is common, and a lack of preparedness to enter practice is shared amongst specialties [18]. Further research should detail the factors which play a role in establishing a good fit between the graduating fellow and the workplace. We found in our study that most of the training programs were already working on helping trainees locate jobs and advising them regarding the decision of what position to choose. Educating training programs on how to better assess “fit” for their trainees may be the first step to take. The current literature suggests a few possible factors to take in consideration. Obtaining appropriate early mentorship at the new institution appears very important. A facility which hires and is open to the development of new faculty, with peer-to-peer teaching and support is innovative and desirable [32]. Development of more robust tools to assess the effectiveness of the training in professionalism, interpersonal, and communication skills should better guide in choosing the effective teaching techniques [8,33]. Finally, defining a desirable outcome from the first employment process and establishing a database linking trainees’ outcomes to training programs could shed light upon the effectiveness of the entire process.

## 5. Conclusions

Young critical care professionals describe a variety of needs in education regarding first employment. Despite the employment of a variety of techniques aiming at improving non-clinical skills, a gap between the demand of their future job and provided training continues to persist. 

## Figures and Tables

**Figure 1 healthcare-07-00074-f001:**
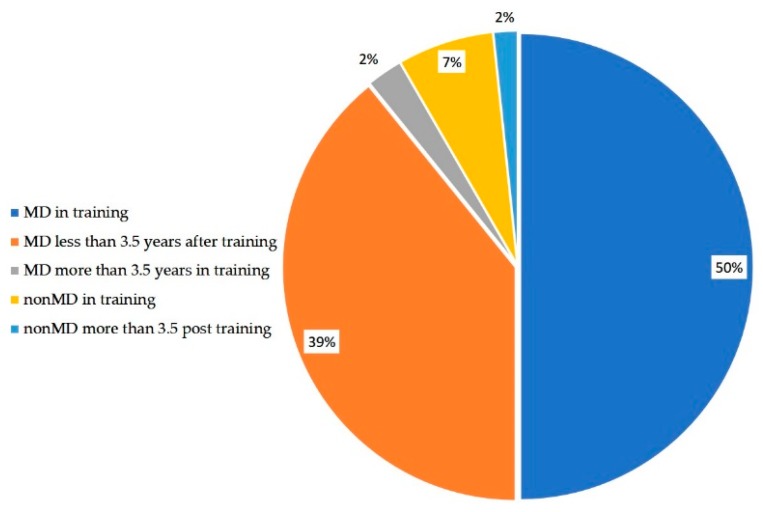
Demographics of people who were currently interviewing or have interviewed during the past 4 years.

**Figure 2 healthcare-07-00074-f002:**
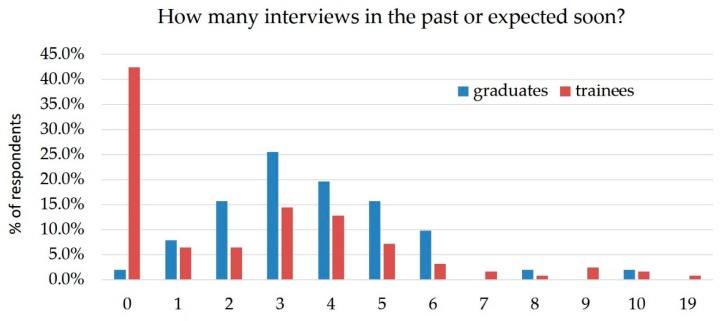
Reported numbers of interviews.

**Figure 3 healthcare-07-00074-f003:**
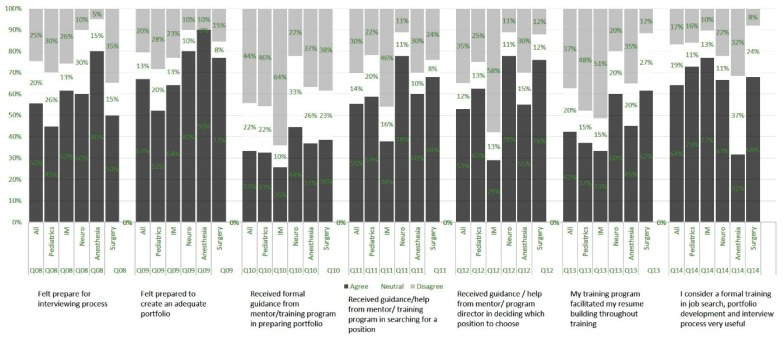
Likert scale responses by specialty.

**Figure 4 healthcare-07-00074-f004:**
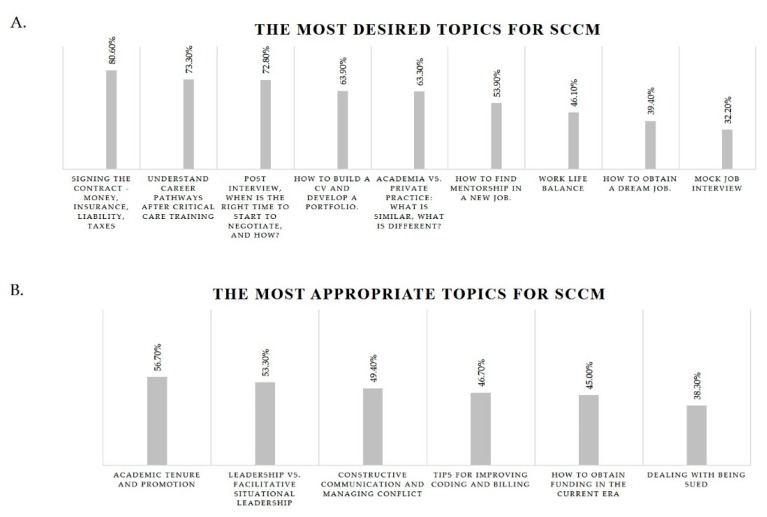
The most desired topics (**A**) and the most appropriate topics (**B**) for discussion at the Society of Critical Care Medicine (SCCM).

**Table 1 healthcare-07-00074-t001:** Respondent demographics expressed as number (percentage).

Group Composition	
**Physician In Training**	112 (62.2%)
Recently Graduated Physician	48 (26.7%)
Non-Physician In Training	13 (7.2%)
Recently Graduated Non-Physician	3 (1.7%)
Other	4 (2.2%)
**Specialties**	
Critical Care Medicine - Pediatrics	66 (37%)
Critical Care Medicine - Internal Medicine	51 (28%)
Critical Care Medicine - Surgery	29 (16%)
Critical Care Medicine - Anesthesiology	21 (12%)
Critical Care Medicine - Neurology	10 (6%)
Other	24 (13%)

**Table 2 healthcare-07-00074-t002:** Job characteristics expressed as number (percentage). Respondents were able to select multiple regions.

Preferred Locations (Could Pick Multiple)	
**Northeast USA**	85 (47%)
Southeast USA	45 (25%)
Northcentral USA	45 (25%)
Southcentral USA	11 (6%)
Northwest USA	40 (22%)
Southwest USA	40 (22%)
Anywhere in USA	22 (12%)
Other Locations	20 (11%)
**Practice Type**	
Full Time Academic	128 (71%)
Full Time Private	24 (13%)
Part Time Academic	16 (9%)
Part Time Private	4 (2%)
Other	8 (5%)

## Data Availability

Data will be made accessible upon reasonable request.

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
