# Peer review of "Self-Assessment of Preparedness among Critical Care Trainees Transitioning from Fellowship to Practice"

_healthcare, 2019, doi:10.3390/healthcare7020074_

Round 1

Reviewer 1 Report

Thank you for addressing the reviewers' comments.

One last comment: In the methods you mention using the "Independent samples T-tests were utilized to compare the distribution of intensive care specialists between medical doctors and other medical providers." This sounds to me like a categorical variable presented as a percentage, where a Chi-Square test would be more appropriate. Also, you presented most of the results as a median and interquartile range. I am guessing this is because the data were skewed. This means that you should have used a Mann-Whitney test not a t-test.

Author Response

One last comment: In the methods you mention using the "Independent samples T-tests were utilized to compare the distribution of intensive care specialists between medical doctors and other medical providers." This sounds to me like a categorical variable presented as a percentage, where a Chi-Square test would be more appropriate. Also, you presented most of the results as a median and interquartile range. I am guessing this is because the data were skewed. This means that you should have used a Mann-Whitney test not a t-test.

    Thank you very much for bringing this important point to our attention. We agree that a Mann-Whitney U test should have been calculated instead of a t-test. We re-calculated the distribution of intensive care specialties, and it was still an insignificant result. We adjusted our methods accordingly to report our use of the test.

This manuscript is a resubmission of an earlier submission. The following is a list of the peer review reports and author responses from that submission.

Round 1

Reviewer 1 Report

This is an interesting study that provides a good insight about the needs of the critical care trainees. Please consider my below comments to improve the quality of the manuscript.

Review

1.       Please change responders to respondents throughout the manuscript

2.       Lines 21-22: these numbers are from a subgroup of respondents who had or secured an interview. Please clarify this information

3.       Line 56: spell out ACGME

4.       Line 57: you did not perform any inferential statistics. This study is descriptive in nature, just state that your objective was to evaluate the perceived preparedness of trainees and recent graduates for the first job transition.

5.        Line 67: change “pertinent hits” to “subjects that meet the study selection criteria.”

6.       Line 67: explain how you were able to identify trainees and recent graduates using the SCCM membership database.

7.       Line 76: I cannot find appendix 1.

8.       Line 76: how did you send the survey? An email with a link to the Google form?

9.       Line 81: add to “test face validity” after the “In-training section.”

10.   Results lines 87-152: please place the number of respondents in the sentence and the percentage between parentheses; also do not start the sentence with number that is not spelled out. For example line 88:  A total of 160 responses were from physicians (89%), and 125 were completing a fellowship (69%).

11.   Line 93-94: this is the only time you refer to inferential statistics. What test did you use to compare the number of intensive care specialists between MDs and non-MDs

12.   Line 108: results show that 47% preferred a job location in the Northeast of the US, but what was the current location of these respondents? Is it possible that most of the survey respondents happened to be from the Northeast?

13.   Line 117: change persons to respondents

14.   Add the number and percentage of respondents who had an interview or going to be interviewed

15.   Line 117-118: why did you decide to ask questions about interviews to this subgroup only? It would make sense if you asked those who already had an interview if they felt prepared or not. Or just asked all respondents. I would think that the perceived preparedness for those interviewing soon will not be different from those who did not get an interview.

16.   Figure 3 is very hard to see. Please spread it out in an entire page, use a better choice of color, and remove the question numbers at the bottom, or just use a table.

17.   Line 139 what is a positive response? Agree or strongly agree or selected from a list? Please rephrase

18.   Lines 139-150: most of the paragraph seems redundant, the information is already in figures 4A and 4B.

19.   Lines 166-167: move to limitations section and expand

20.   Lines 171-175: this part is unrelated to the study and adds nothing to the discussion

21.   Line 181: an international audience may not be familiar with career pathways

22.   Lines 185-187: I am not sure where you presented these results.

23.   There is no discussion of the limitations of the study, including the extremely low response rate. You have to make an argument as to why you believe that the sample is representative of your target population. It is possible that you received responses mostly from individuals who are having difficulty finding a job or of lower performance. You need to convince the reader the results are valid and not biased. Also, the survey measured the perceptions of trainees and recent graduates; please discuss the implications of collecting subjective versus objective data and what future research is needed. Moreover, there is minimal discussion of what has been introduced in training fellows over the past few years to help them find a job, and if there was an evidence of improvement in the study results linked to any of these changes (lines 185-186 see my comments above).

Author Response

Response to Reviewer #2 Comments

The manuscript addresses a critical topic of interest and relevance to critical care community especially in the context of increasing 'burn out' in critical care community. The development of career enhancing  'soft skills' is vital. The continued lack of structured training during fellowship to address these lacunae remains a critical issue. I believe the manuscript addresses, highlights these issues; which are very relevant in current health care environment towards training a future generation of  'resilient' critical care practitioners.

Thank you very much for your kind words. We hope that you consider the changes to our manuscript adequate for publication.

However, I have a few major queries:

a) The manuscript pertains to data collected in 2013-2014. Although the findings of the study will still be applicable and probably more relevant in 2019, one would query the authors on publication of 2013-2014 data in 2019 and reasons for the delayed publication,

Thank you for your concern. Regarding delayed publication, much of the delay is due to the extensive reviewing period from the previous journal we submitted our manuscript.

b) I have concerns regarding the general applicability of study findings to the broad critical care community.

The study responders’ sample does not seem to be reflective of the current critical care work force

- I do not see any representation of pulmonary critical care - the most common training pathway in critical care internal medicine. Pulmonary critical care accounts for 50% of fellows of entire internal medicine critical care work force which comprises the vast majority (71%) of critical care work force in USA.1,2 .

Does Critical Care Medicine Internal Medicine (28%) include the pulmonary critical care pathway?

Yes, this includes both pathways. We added a sentence to the results section that explicitly mentions this point.

- The majority of responders (37%) were Critical Care Medicine Pediatrics, which again is not reflective of current critical care community.

            We addressed this concern as a limitation of our study.

- Although I understand the authors have little say or control over who responds to the survey, it is statistically unusual to have minimal representation from the most common training pathway in critical care (internal medicine)

            While critical care internal medicine was certainly under-represented,

- This finding may raise the issue of responder’s bias or a methodology issue in not reaching out to the target audience for the survey responders.

c) The response rate of 6% is on the lower side and may explain to some extent the limitation of the study findings.

We agree and added this point to our limitations. However, we also cited a previous study using the SCCM database that obtained a similar response rate.

d) Recall bias: from responders who finished their training (within 3 years) accounting for 27.1% would be a valid concern and limitation.

Thank you very much for your suggestion. We hope that our limitations section within the discussion satisfies your concerns.

I suggest incorporating, addressing these issues in discussion and limitations of the study.

References:

1. Halpern NA, Pastores SM, Oropello JM, Kvetan V. Critical care medicine in the United States: addressing the intensivist shortage and image of the specialty. Crit Care Med. 2013 Dec;41(12):2754-61.

2. ABMS 2011 Certificate Statistics. Available at: http://www.abms.org.

Reviewer 2 Report

The manuscript addresses a critical topic of interest and relevance to critical care community especially in the context of increasing 'burn out' in critical care community.

The development of career enhancing  'soft skills' is vital. The continued lack of structured training during fellowship to address these lacunae remains a critical issue.

I believe the manuscript addresses, highlights these issues; which are very relevant in current health care environment towards training a future generation of  'resilient' critical care practitioners.

However, I have a few major queries:

a) The manuscript pertains to data collected in 2013-2014.

Although the findings of the study will still be applicable and probably more relavant in 2019, one would query the authors on publication of 2013-2014 data in 2019 and reasons for the delayed publication,

b) I have concerns regarding the general applicability of study findings to the broad critical care community.

The study responders sample does not seem to be reflective of the current critical care work force

- I do not see any representation of pulmonary critical care - the most common training pathway in critical care internal medicine. Pulmonary critical care accounts for 50% of fellows of entire internal medicine critical care work force which comprises the vast majority (71%) of critical care work force in USA.1,2 .

Does Critical Care Medicine Internal Medicine (28%) include the pulmonary critical care pathway?

- The majority of responders (37%) were Critical Care Medicine Pediatrics, which again is not reflective of current critical care community.

- Although,I understand the authors have little say or control over who responds to the survey, it is statistically unusual to have minimal  representation from the most common training pathway in critical care (internal medicine)

- This finding may raise the issue of responders bias or a methodology  issue in not reaching out to the target audience for the survey responders.

c) The response rate of 6% is on the lower side and may explain to some extent the limitation of the study findings.

d) Recall bias: from responders who finished their training (within 3 years) accounting for 27.1%  would be a valid concern and limitation.

I suggest incorporating, addressing these issues in discussion and limitations of the study.

References:

1. Halpern NA, Pastores SM, Oropello JM, Kvetan V.

Critical care medicine in the United States: addressing the intensivist shortage and image of the specialty.

Crit Care Med. 2013 Dec;41(12):2754-61.

2. ABMS 2011 Certificate Statistics. Available at: http://www.abms.org.

Author Response

This is an interesting study that provides a good insight about the needs of the critical care trainees. Please consider my below comments to improve the quality of the manuscript.

­Thank you very much for the kind words. We hope that you consider our changes adequate for publication.

1.       Please change responders to respondents throughout the manuscript

All references to “responders” have been changed to “respondents”.

2.       Lines 21-22: these numbers are from a subgroup of respondents who had or secured an interview. Please clarify this information

Thank you for this point. We clarified this point within the abstract.

3.       Line 56: spell out ACGME

Completed.

4.       Line 57: you did not perform any inferential statistics. This study is descriptive in nature, just state that your objective was to evaluate the perceived preparedness of trainees and recent graduates for the first job transition.

Thank you for the suggestion. We changed the sentence to read, “…our study sought to evaluate the perceived preparedness of trainees and recent graduates for their first job transition”.

5.       Line 67: change “pertinent hits” to “subjects that meet the study selection criteria.”

Completed.

6.       Line 67: explain how you were able to identify trainees and recent graduates using the SCCM membership database.

Thank you. We added the following information, “The database of SCCM membership was searched for subjects that met the study selection criteria. The SCCM database contains self-reported biographical information that users provide during the SCCM membership process.”.

7.       Line 76: I cannot find appendix 1.

Sorry for the inconvenience. We added the survey in the Appendix on this resubmission.

8.       Line 76: how did you send the survey? An email with a link to the Google form?

We added the following sentence, “The survey was deployed twice via email containing links to the survey site during the period of December 2013 to January 2014.”

9.       Line 81: add to “test face validity” after the “In-training section.”

10.   Results lines 87-152: please place the number of respondents in the sentence and the percentage between parentheses; also do not start the sentence with number that is not spelled out. For example line 88:  A total of 160 responses were from physicians (89%), and 125 were completing a fellowship (69%).

Completed.

11.   Line 93-94: this is the only time you refer to inferential statistics. What test did you use to compare the number of intensive care specialists between MDs and non-MDs

Thank you. We added the following sentence in the methods section, “Independent samples T-tests were utilized to compare the distribution of intensive care specialists between medical doctors and other medical providers.”

12.   Line 108: results show that 47% preferred a job location in the Northeast of the US, but what was the current location of these respondents? Is it possible that most of the survey respondents happened to be from the Northeast?

Thank you for raising this point. SCCM provided us with an anonymous sample that is intended to be nationally representative. Since our response rate was 6%, it is possible that there is bias regarding location preferences.

13.   Line 117: change persons to respondents

Completed.

14.   Add the number and percentage of respondents who had an interview or going to be interviewed

Completed.

15.   Line 117-118: why did you decide to ask questions about interviews to this subgroup only? It would make sense if you asked those who already had an interview if they felt prepared or not. Or just asked all respondents. I would think that the perceived preparedness for those interviewing soon will not be different from those who did not get an interview.

We decided to evaluate the perceived preparedness among those that received an interview, because we believe that this data provided us with the most accurate assessment of preparedness. While it is possible that the perceived preparedness would not differ between groups, we did not design our survey to capture that data.

16.   Figure 3 is very hard to see. Please spread it out in an entire page, use a better choice of color, and remove the question numbers at the bottom, or just use a table.

Thank you. We enlarged the figure to take up an entire page.

17.   Line 139 what is a positive response? Agree or strongly agree or selected from a list? Please rephrase

Thank you for the suggestion. We wrote the following, “positive response (defined as agree or strongly agree on the Likert scale)

18.   Lines 139-150: most of the paragraph seems redundant, the information is already in figures 4A and 4B.

Thank you very much for your suggestion. We trimmed some of this section down, but we believe that

19.   Lines 166-167: move to limitations section and expand

Thank you for your suggestion. We moved it to a paragraph discussing limitations and expanded upon it.

20.   Lines 171-175: this part is unrelated to the study and adds nothing to the discussion

Thank you for your suggestion. We deleted this portion from the discussion.

21.   Line 181: an international audience may not be familiar with career pathways

Thank you very much for raising this important point. We decided to remove this piece to avoid confusion.

22.   Lines 185-187: I am not sure where you presented these results.

While our data suggests that the majority of trainees agreed that they received guidance from a mentor/program in searching for a position and deciding which position to choose, we made the inference that inference as written in lines 185-187. We re-wrote this piece to reflect a more conservative view of the data.

23.   There is no discussion of the limitations of the study, including the extremely low response rate. You have to make an argument as to why you believe that the sample is representative of your target population. It is possible that you received responses mostly from individuals who are having difficulty finding a job or of lower performance. You need to convince the reader the results are valid and not biased. Also, the survey measured the perceptions of trainees and recent graduates; please discuss the implications of collecting subjective versus objective data and what future research is needed. Moreover, there is minimal discussion of what has been introduced in training fellows over the past few years to help them find a job, and if there was an evidence of improvement in the study results linked to any of these changes (lines 185-186 see my comments above).

Thank you very much for your suggestions. We have re-written much of our discussion, and we hope that you find that these changes adequately addressed your concerns.
